# The Role of circTmeff-1 in Morphine Addiction Memory of Mice

**DOI:** 10.3390/cells12151985

**Published:** 2023-08-01

**Authors:** Hailei Yu, Boyang Wen, Yun Lu, Bing Xie, Feng Yu, Minglong Zhang, Chunling Ma, Bin Cong, Di Wen, Haitao Bi

**Affiliations:** 1Hebei Key Laboratory of Forensic Medicine, Collaborative Innovation Center of Forensic Medical Molecular Identification, Research Unit of Digestive Tract Microecosystem Pharmacology and Toxicology, College of Forensic Medicine, Hebei Medical University, Chinese Academy of Medical Sciences, Shijiazhuang 050000, China; 18232587910@163.com (H.Y.); boyangwen@126.com (B.W.); 15226523170@163.com (Y.L.); queenie06@163.com (B.X.); fengyu1405@163.com (F.Y.); chunlingma@126.com (C.M.); hbydbincong@126.com (B.C.); 2Department of Biogenetics, Qiqihar Medical University, Qiqihar 161000, China; jylinyao@163.com

**Keywords:** circRNAs, morphine, conditional place preference, addiction memory, nucleus accumbens

## Abstract

In addition to the essential pharmacological effects of opioids, situational cues associated with drug addiction memory are key triggers for drug seeking. CircRNAs, an emerging hotspot regulator in crown genetics, play an important role in central nervous system-related diseases. However, the internal mediating mechanism of circRNAs in the field of drug reward and addiction memory remains unknown. Here, we trained mice on a conditional place preference (CPP) model and collected nucleus accumbens (NAc) tissues from day 1 (T0) and day 8 (T1) for high-throughput RNA sequencing. QRT-PCR analysis revealed that circTmeff-1 was highly expressed in the NAc core but not in the NAc shell, suggesting that it plays a role in addiction memory formation. Meanwhile, the down-regulation of circTmeff-1 by adeno-associated viruses in the NAc core or shell could inhibit the morphine CPP scores. Subsequently, the GO and KEGG analyses indicated that circTmeff-1 might regulate the addiction memory via the MAPK and AMPK pathways. These findings suggest that circTmeff-1 in NAc plays a crucial role in morphine-dependent memory formation.

## 1. Introduction

Drug addiction is a chronic recurrent encephalopathy mainly based on the neural adaptation or neuroplasticity of the central nervous system [1]. Opioids—a primary addictive drug—mainly induce mental and physical dependence in animals by activating µ receptors in G protein-coupled receptors [2]. The mortality rate associated with opioid abuse has increased by 500% per cent over the past two decades [3]. Therefore, it is crucial and urgent to explore the mechanism and therapeutic targets of opioid addiction [4]. Conditioned place preference (CPP), based on the classical Pavlovian conditioned reflex principle, is a classical experimental model for evaluating the effect of drug reward and psychological dependence [5]. Drug reward and environmental cues can be associated with the paradigm [6]. Substantial research has revealed that nucleus accumbens (NAc) is closely associated with drug-induced reward, psychological craving, reinforcement, and other effects [7,8]. Over the years, several molecules have been proven to control addictive memory, such as the increase in matrix metalloproteinase (MMP)-9 in the midbrain and the elevation of tissue plasminogen activator (tPA) in the frontal cortex and NAc [9,10]. However, little is known about the role of non-coding RNAs that may be critical in the development of context-induced drug seeking.

Circular RNAs (circRNAs) are a class of noncoding RNA molecules that form covalent ring structures without free 5′ and 3′ ends via a non-canonical splicing event known as backsplicing [11]. There are several types of circRNAs, such as exon–intron circRNA and exonic circRNA produced by backsplicing [12]. In addition, exon skipping occurs in the final mRNA product, and the internal backsplicing of the exon can also form exonic circRNA [13], while canonical linear splicing of introns may also lead to the generation of circular intronic RNAs [14]. Recent studies have suggested that circRNAs may regulate target mRNAs by acting as adsorption sponges for microRNAs (miRNAs) [15]. For instance, circRNA_002381 exerts a sponge effect on miR-138 to partially modulate mRNAs associated with drug addiction [16]. The length and type of circRNAs have spatiotemporal specificity of cell types, tissues, and diseases [17,18]. Furthermore, circRNAs exhibit peak dynamic expression in mammalian neuronal tissues and cells [19] and play an essential regulatory role in nervous system diseases such as acute CNS insults, schizophrenia, and glioblastoma [20,21,22]. However, the expression of circRNAs in morphine-induced drug addiction memory has not been studied well.

CircTmeff-1 (mmu_circ_0011663), with a length of 706 BP, derives from the Tmeff-1 gene on chromosome 4 [23]. Tmeff-1 is a putative transmembrane protein that regulates the function of central nervous system [24]. T. Arano et al. have identified that Tmeff-1 has an affinity for the addiction-related protein addicsin, which exhibits high expression in brain tissue following morphine administration [25]. Our previous study found that circTmeff-1 in the NAc plays an essential regulatory role in incubating context-induced morphine craving [23], but it remains unclear how circTmeff-1 affects morphine addiction memory. Interestingly, the sequencing results revealed an up-regulation in the expression of circTmeff-1 following morphine-dependent memory expression. This study investigated the expression and the possible mechanism of circTmeff-1 during morphine-induced addiction memory.

## 2. Materials and Methods

### 2.1. Animals

A total of 224 male C57BL/6 N mice (20–25 g) were obtained from Liaoning Changsheng Co., Ltd. (Liaoning, China). The animals were housed at constant temperature (21 °C ± 2 °C) and humidity (approximately 60–65%), with a 12 h light/dark cycle (lights on at 8:00 a.m.). The experiments were conducted according to the National Institutes of Health Guide for the Care and Use of Laboratory Animals.

### 2.2. Drugs

Morphine hydrochloride (Qinghai, China) and cocaine (Qinghai, China) were diluted to 10 mL/kg body weight with normal saline for injection. Methamphetamine (purity >95%, supplied by the Public Security Bureau of Beijing Municipality, Beijing, China) concentrations was adjusted to an appropriate injection dose of 1 mL/kg body weight with saline. High-sugar foods were homemade.

### 2.3. Behavior Test

#### 2.3.1. Conditioned Place Preference

The conditioned place preference (CPP) evaluated the effect of circTmeff-1 on the memory of morphine addiction [26]. In this experiment, a two-box unbiased design was adopted. The two compartments were separated by a baffle with a small door (JLBeh Soft-tech Co., Ltd., Shanghai, China) and differed significantly in tactile and visual terms. The CPP scores were obtained by subtracting the time spent in the morphine-paired compartment from the time spent in the saline-paired compartment. On the first day (T0), the small door was opened, and the mice were allowed to explore freely in the box for 15 min. The time spent on both sides was recorded, and the CPP score was calculated. The mice with CPP greater than 150 s were screened out. From the second day to the seventh day, the channels of both compartments were closed, and the mice were scheduled to receive injections at 9 A.M and 3 P.M every day. The mice administered with morphine were placed on the drug-administered side for 45 min. Another half-day, the same mouse was injected with saline and placed on the other side. To prevent the mice from associating the time of day with drug administration, we administered morphine in the morning on days 2, 4, and 6 and in the afternoon on days 1, 3, and 5. On day 8 (T1), the wicket was opened, post-tests were conducted on the mice without injection, and the mice were allowed to move freely in the device for 15 min, as was the case with T0.

The cocaine/methamphetamine/food CPP was consistent with the aforementioned strategy. In food CPP training, we placed high-glycemic food alongside the drug as a reward.

#### 2.3.2. Light–Dark Box

In the light–dark box, bright light illuminated one-third of the dark box and the entire light box. The partition wall between the two boxes had a 7.5 cm × 7.5 cm door passageway for animals (JLBeh Soft-tech Co., Ltd., Shanghai, China). The mice were placed in the center of the open box, and the passage and retention times in the open box and the dark box within 5 min were recorded.

#### 2.3.3. Locomotion Test

The mice were placed in the center of the open field (25 cm × 25 cm; JLBeh Soft-tech Co., Ltd., Shanghai, China), and their movement track and distance over a 30 min period were recorded using the SMART video tracking system (Panlab Technology for Bioresearch, Barcelona, Spain).

#### 2.3.4. Barnes Maze

The Barnes maze was a 24-hole maze with removable escape boxes placed under the target holes (JLBeh Soft-tech Co., Ltd., Shanghai, China). The day before the experiment, each mouse was placed in the target box through the target hole for 5 min. After 5 days of escape training, the escape box was removed on day 6 of the experiment, and the incubation period of mice reaching the target hole and the number of errors per animal were recorded using the SMART video tracking system (Panlab Technology for Bioresearch, Barcelona, Spain).

### 2.4. RNA Sequencing

The RNA sequencing methodology utilized in this study was analogous to that employed in our previous publication [23]. The mice’s NAc tissues (approximately 600 μm) were extracted 30 min after the CPP test (T0 and T1) *(n* = 3 samples per group). RNA was extracted from NAc using TRIzol reagent (Takara, Japan). After verifying that the RNA quality meets the library construction requirements, a total of 1.5 μg RNA per sample was utilized as input material for rRNA depletion utilizing the Ribo-Zero rRNA Removal Kit (Epicentre, Madison, WI, USA). The NEBNext^®^ UltraTM Directional RNA Library Prep Kit for Illumina^®^ (NEB, Ipswich, MA, USA) was utilized to prepare sequencing libraries, which were subsequently sequenced on an Illumina HiSeq X sequencer (Illumina, San Diego, CA, USA). To identify and quantify circRNAs, the raw reads from RNA sequencing data were subjected to processing using in-house Perl scripts, and high-quality reads were obtained. Each sample’s remaining reads were mapped to the mice reference genome using GRCm38. The circRNAs were identified as those with at least one unique backspliced site. Subsequently, HTseq was utilized to compute the count and RPM values of circRNAs in each sample. Finally, we used DESeq in R with default parameters to analyze differential expression of circRNAs.

### 2.5. Quantitative Real-Time PCR

About 30 min following the T0 and T1 CPP tests, approximately 600 um sections of the NAc tissues were collected by micro punch (BT1.0, Harris, Melbourne, FL, USA). TRIzol reagent (Takara, Kusatsu, Japan) was used to extract total RNA from NAc, and 500 ng of RNA was reverse-transcribed into cDNA using the PrimeScript RT reagent Kit (TaKaRa, Kusatsu, Japan). Amplification and detection were performed in the quantitative PCR instrument (Applied Biosystems, Waltham, MA, USA) using the real-time fluorescence PCR kit (TaKaRa, Beijing, China). The protocol of reaction conditions is shown in Table 1. All reactions were performed on a QuantStudio Flex (Applied Biosystems) in a volume of 20 μL.

In this study, the GAPDH mRNA served as the internal reference, and 2-ΔΔCt was used to calculate the expression levels of circRNA. The primers are as follows: circTmeff-1:
upstream primer 5′-CAGAAGGCCTCTTG TAGAATTA-3′, downstream primer 5′-CATTTCAAACCATCTCCATCTTC-3′;
circDcun1d4:
upstream primer 5′-GTGGTTTTATGAATATGCAGACTT-3′,downstream primer 5′-ACGTCTTCTGTCAGATTCAG-3′; 
GAPDH:
upstream primer 5′-GCTCGTCGTCGACAACGGCTG-3′, downstream primer 5′-CAAACATGAT CTGGGTCATCTTTC-3′.


### 2.6. CircRNA Target Prediction

MiRanda (animal), RNAhybrid (animal), and TargetFinder (plant) tools were used to predict target miRNA. The input files were circTmeff-1 FASTA sequences files, and 33 overlapping target micro RNAs (miRNAs) were obtained among the three aforementioned results.

### 2.7. Functional Analysis

The potential functions of the 33 overlapping target miRNAs were predicted using Gene Ontology (GO, http://geneontology.org/, accessed on 15 March 2022) and Kyoto Encyclopedia Genes and Genomes (KEGG, http://www.kegg.jp/, accessed on 16 March 2022) analyses. The GO analysis was performed to identify relevant genes through biological processes, cellular components, and molecular functions, whereas KEGG was performed to identify the relevant pathway functions involved in circTmeff-1 from large-scale molecular data sets. 

### 2.8. RNA Fluorescence In Situ Hybridization

The mice were sacrificed within 1 h after the T1 test, and their NAc tissues were fixed in 4% formaldehyde for 48 h. Approximately 30 μm-thick brain slices of the coronal NAc layer (bregma + 0.74 mm to + 1.7 mm) were cut using an Oscillating microtome (Leica VT1000 S, Wetzlar, Germany.) The brain slices were drilled in 3% Triton-X 100 (Thermo Fisher Scientific, Waltham, MA, USA) and chilled for 1 h at 4 °C. After removing the brain slices from the refrigerator, the slices were then sealed with a prehybridization solution (Beijing Nuowei Biotechnology Co., LTD., Beijing, China) at 37 °C for 1 h. The sealed slices were incubated with a circTmeff-1 probe (Integrated Biotech Solutions, Shanghai, China) overnight at 37 °C. A confocal microscope (Leica SP8, Wetzlar, Germany) was used on the second day after sealing with DAPI (Beijing Solarbio, Beijing, China).

### 2.9. Adeno-Associated Virus Injection

The expression of circTmeff-1 in the NAc core and shell was modulated using the adeno-associated virus (AAV) of the Tet-off system. The AAV vector was constructed by Wuhan Shumi Brain Science and Technology Co., LTD (Wuhan, China) (for details, see Table 2). 

Doxycycline (DOX; 50 mg/L) was prepared with drinking water before the AAV injection. Specific and auxiliary viruses were mixed in a ratio of 1:1. The mice were anesthetized with isoflurane and then immobilized in a stereotactic framework (NeuroStar, Germany). The output rate of the injection pump (Hamilton, Lancaster, PA, USA) was adjusted to 0.04 μL/min; then, 0.2 μL of the mixed virus was injected on each side of the framework. The remaining steps are detailed in our previous paper [23].

### 2.10. Statistics

Behavioral data were tested using the paired t-test or one-way analysis of variance. The gene (circTmeff-1, circDcun1d4) data were tested using the unpaired t test. The experimental data were statistically analyzed using the SPSS 21.0 software, and the independent sample t-test was used to compare differences between the two groups. One-way analysis of variance was used for three or more groups, and the Levene test was used to determine the homogeneity of variance. *p* < 0.05 indicates statistical significance. All data graphs in the experiment were plotted using GraphPad 8.2 software, and the data are expressed as mean ± S.E.M.

## 3. Results

### 3.1. CircRNAs Expression Profile Changed in the Nucleus Accumbens of Mice with Morphine-Addicted Memory Formation

To study the changes in circRNA expression level associated with morphine addictive memory in the NAc, we administered four groups of mice with normal saline or morphine (1, 5, or 10 mg/kg), and morphine CPP was tested on day 1 (T0) and day 8 (T1) (Figure 1A(a)). The post hoc tests uncovered that CPP scores were significantly increased by 5 and 10 mg/kg doses of morphine (Figure 1A(b)). Accordingly, circRNA sequencing was performed using NAc tissue samples from the mice treated with 10 mg/kg of morphine after T0 and T1. The heatmap (Figure 1B) revealed that two circRNAs increased, while three decreased. The RNA sequencing data were verified using quantitative real-time PCR (qRT-PCR) (Figure 1C), and the analysis results indicated a more significant increase in circTmeff-1. We will only focus on circTmeff-1 in this story, although we also find other interesting targets. After mixing with RNase R, qRT-PCR analysis revealed that circTmeff-1 was stabler than its linear transcript (Figure 1D). Moreover, the head–tail splicing of circTmeff-1 was verified using divergent primers (Figure 1E). The annular structure of circTmeff-1 was further proved. Fluorescence in situ hybridization revealed increased circTmeff-1 expression on T1 (Figure 1F).

### 3.2. The Expression of circTmeff-1 Increased in the NAc Core in Mice with Morphine-Addicted Memory Formation

As NAc core and shell play different roles in addiction, the expression level of circTmeff-1 was detected. As shown in Figure 2A, the expression of circTmeff-1 in the NAc core was significantly higher at T1 compared with T0, and no changes were observed in the shell. The effect of CPP on circTmeff-1 expression in mice was detected on days 1, 3, 5, and 7. The data revealed a gradual increase in circTmeff-1 expression among conditioned mice with CPP training, which was significantly higher than those without CPP on days 5 and 7. (Figure 2B). These findings suggest that the circTmeff-1 in the NAc core may contribute to morphine addiction memory.

The expression of circTmeff-1 in NAc during addiction induced by other drugs or substance was also detected. Like the previous regime, the mice were trained with cocaine, methamphetamine, and food (Figure 2C(a),D(a),E(a)). The expression of circTmeff-1 in NAc was then studied. Stable addictive memory was formed with 10 mg/kg (*p* < 0.001) of cocaine (Figure 2C(b)). Addictive memory was also formed with 1 mg/kg (*p* < 0.001) of methamphetamine (Figure 2D(b)). Food reward could contribute to the significant increase in the CPP score (Figure 2E(b)). Substantial up-regulation of circTmeff-1 expression in the NAc core was observed in both cocaine (10 mg/kg) and methamphetamine (1 mg/kg) groups (Figure 2C(c),D(c)). However, there is no alteration in circTmeff-1 in the NAc core and shell in the food group (Figure 2E(c)). These findings indicate that circTemff-1 in the NAc core involves drug addiction rather than natural reward.

### 3.3. CircTmeff-1 in NAc Core and Shell Plays a Role in Morphine-Dependent Memory Formation

To investigate the role of circTmeff-1 in morphine-induced addiction memory, adeno-associated virus (AAV) vectors carrying circTmeff-1 or shcircTmeff-1 were employed to modulate its expression specifically in the NAc core and shell. The mice began to drink water containing doxycycline (50 mg/L) 21 days before T0, and viruses were microinjected into the distinct areas of NAc 20 days before T0, as shown in Figure 3A. 

The viral down-regulation of circTmeff-1 significantly decreased CPP scores in both NAc core (*p* < 0.001) and shell (*p* < 0.001) on T1 compared with control or scramble group (Figure 3B(b)). However, the over-expression of circTmeff-1 in the NAc had no significant effect. These findings suggest that circTmeff-1 in the NAc core and shell may participate in the process of morphine addiction memory. 

### 3.4. CircTmeff-1 in NAc Core and Shell Did Not Affect Motor Activity, Anxiety-Like Behavior, and Spatial Memory of Mice

Since AAV may impair the spatial memory of mice, the Barnes maze test was conducted (Figure 4A(a)). The time spent by mice to find the target hole was not affected by microinjection of shcircTmeff-1 in the NAc. As the function of various circTmeff-1 expressions on CPP scores might be affected by the motor ability of mice, we administered a spontaneous locomotion test on T1. The analysis revealed no locomotion change in the mice with over-expression or with down-regulation of circTmeff-1 (Figure 4B(a),C(a)). Given that anxiety status might also affect CPP scores [27], the light–dark box test was conducted. The statistical analysis demonstrated that the retention time of mice injected with adenovirus in the bright box revealed no difference, whether in core or shell (Figure 4B(b),C(b)).

### 3.5. GO and KEGG Pathway Analysis

To better understand the mechanism of circTmeff-1′s involvement in the formation of morphine addiction, 33 overlapping target micro RNAs (miRNAs) of circTmeff-1 were obtained (Figure 5A). GO and KEGG enrichment analyses of the miRNAs were subsequently performed. Figure 5B–D list the top ten GO high-enrichment items. The items that were rich and related to biological processes, cell component, and molecular function were “small GTPase mediated signal transduction”, “extracellular region”, and “GTP binding”, respectively. According to the KEGG analysis (Figure 5E), we found two pathways that are relatively related to drug addiction, namely, “MAPK signaling pathway” [28] and “AMPK signaling pathway” [29].

## 4. Discussion

In our prior investigation, it was discovered that circTmeff-1 located in the nucleus accumbens (NAc) core promotes the incubation of morphine craving in mice by acting as a sponge for miR-541/miR-6934 [23]. The present study identified that the escalation of circTmeff-1 in the NAc core and shell during the construction of the morphine-induced addiction memory model is crucial.

The expression of circRNAs, associated with the formation or modification of neural synapses, is remarkably altered after cocaine-induced CPP [30]. In this study, circTmeff-1 increased significantly with CPP morphine training (Figure 1C). Morphine administration can cause changes in the content of many molecular compounds within the body, such as L-tryptophan [31]. To explore whether the expression of circTmeff-1 changes during the establishment and expression of morphine CPP, we examined the expression of circTmeff-1 when morphine was administered alone without training. The results demonstrated no significant change in circTmeff-1 (Figure 2B). Therefore, we speculate that circTmeff-1 may contribute to the formation and expression of morphine addiction memory, also known as initial drug reinforcement [32]. 

The structure of the NAc is heterogeneous not only in anatomy but also in function [33]. Projections into the NAc shell and core are parallel and separate in most brain regions, such as dorsal structures into the nucleus and ventral structures into the shell [34]. Glutamatergic transmission from the PFC and parahippocampal structure to the NAc core is crucial in the context-induced seeking reinforcement [35,36]. This finding might explain why circTmeff-1 in the NAc core increased on T1. Meanwhile, the projection from the hippocampal to the NAc shell is also essential in establishing context connections and behavior in response to psychostimulating drugs [37]. Furthermore, the NAc shell preferentially contributes more to the degree of the behavior of the early cue-induced response [38]. Notably, hippocampal LTP was impaired during morphine CPP expression [39]. Our findings indicate that although there was no significant increase in the expression of circTmeff-1 in the NAc shell on T1 (Figure 2A), down-regulation of circTmeff-1 in this region resulted in inhibited CPP scores (Figure 3B(b)). We speculate that normal level of circTmeff-1 in the NAc shell is essential in the process of memory formation associated with morphine addiction. Otherwise, the NAc core and shell also play different roles in the reinstatement and incubation of some drugs craving and behaviors induced by narcotics [37,40]. We also demonstrated that circTmeff-1 in the NAc core increased after CPP expression in cocaine and methamphetamine. Furthermore, there is some overlap between the brain regions and biochemical reactions involved in natural and drug rewards, but they are not the same [41]. Palatable food, such as fried food, could enhance the motivation for food by adjusting the function of NAc [42]. This is also known as food addiction [43]. To confirm whether the increase in circTmeff-1 is a specific response to the drug reward effect or a non-specific physiological response, we administered fried food CPP. The results confirmed that circTmeff-1 was not involved in food reward, indicating the drug specificity of circTmeff-1, which can be involved in addiction processes with certain drugs but not in natural reward.

The Barnes maze test was administered to ensure that the injection of circTmeff-1-AAV/shcircTmeff-1-AAV did not affect the spatial memory of mice [44]. Considering that the changes in volume, complex internal structure, and function of NAc are closely correlated with anxiety [45], we tested whether the anxiety behavior of mice could be affected by the injection of adenovirus in the two subregions of NAc. Simultaneously, we also confirmed that the injection could not affect the motor behavior of mice.

As a seed set of non-coding RNAs in epigenetics, circRNAs are abundant and have novel mechanisms. They can regulate parental gene transcription in the nucleus [46], and the function of miRNAs can also be affected by competing with miRNA-targeted mRNAs for binding sites [47]. In addition to the key mechanisms that are the focus of current research, circRNAs can also enhance the interaction between two proteins by binding to two or one of them [48]. After serum deprivation stress, circACC1 increases and binds to beta and gamma regulatory subunits of AMPK to promote their interaction, thus promoting the stability and activity of the AMPK holoenzyme [49]. We further analyzed the function and potential pathways associated with circTmeff-1 through GO and KEGG enrichment. The first biological process was “small GTPase mediated signal transduction”, and the highly enriched biological function was “GTP binding”, and both were closely related to the same highly enriched MAPK signaling pathway. The MAPK signaling pathway plays a role in opioid addiction [50], the most relevant of which is the ERK subfamily [51]. Because of the high enrichment of GTP activities, we speculate that circTmeff-1 may inhibit the small GTPase activating protein GAP (promoting the hydrolysis of GTP to GDP), resulting in the formation of more Ras-GTP complexes, and the MEK signaling cascade where the complexes are located would increase the phosphorylation of ERK. Finally, the downstream targets CREB and Elk-1 are more strongly activated, enhancing addictive memories [52,53]. This pathway also plays a vital role in behavioral sensitization [54] and drug cravings [55]. In addition, it is noted that a highly enriched AMPK signaling pathway regulates cellular energy homeostasis [29]. Wyatt B demonstrated that AMPK-controlled energy metabolism is essential for synaptic plasticity, and that activation of AMPK can inhibit the mTOR pathway, thus affecting LTP formation [56]. 

The work conducted in this experiment was far from enough. First, our experiments were conducted solely on male mice. The dissimilarity between animal and human drug abuse with respect to sex is apparent [57]. The density of morphine receptor expression in rats of varying sexes is intrinsically distinct [58], and the gene expression of opioid receptor and the metabolism of opioids in vivo are intimately linked to the exposure of sex hormones [59,60]. Furthermore, neural activity in the nucleus accumbens is primarily affected by dopamine, and a human study has demonstrated differential dopamine production in response to amphetamine therapy between sexes [61]. Additionally, animal studies have demonstrated that estradiol amplifies the intensity of cocaine self-administration and dopamine production induced by cocaine in ovariectomized rats, but not in males [62,63]. In order to mitigate the impact of sex hormone fluctuations during estrus, male mice were utilized. Nevertheless, an investigation into the distinctive sex disparities in morphine-addicted memory formation and circTmeff-1 expression during this phase is intended, as it would hold considerable importance. Next, we will validate the relevant proteins from the functional analysis to further elucidate the downstream target proteins through which circTmeff-1 may regulate morphine addiction memory.

## 5. Conclusions

In conclusion, circTmeff-1 plays an essential role in background-induced morphine addiction, and circTmeff-1 in the NAc could be further investigated as a therapeutic target for addiction prevention.

## Figures and Tables

**Figure 1 cells-12-01985-f001:**
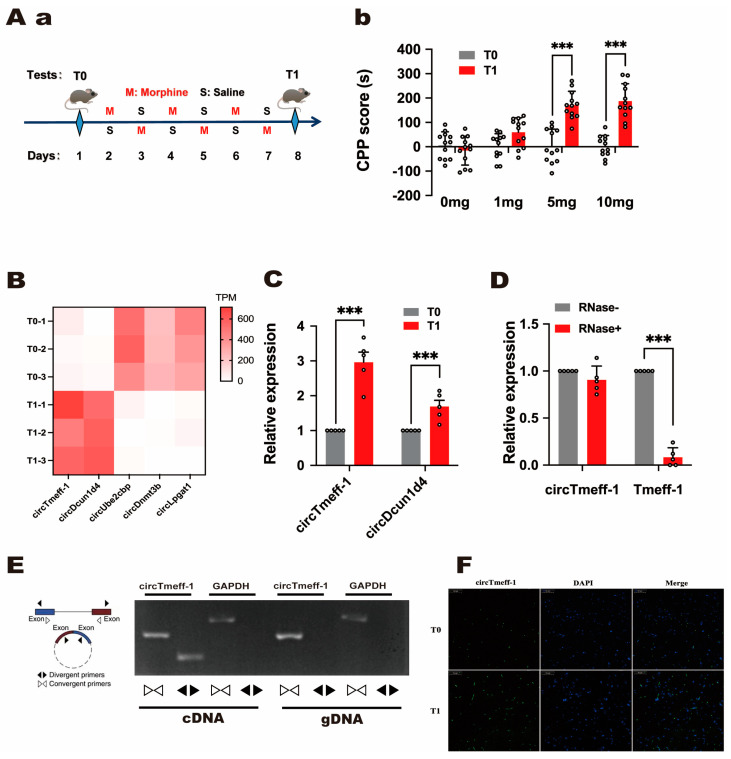
CircTmeff-1 expression increases during context-induced morphine seeking. (**A**(**a**)): a schematic diagram of the timeline of the morphine CPP training program. (**A**(**b**)): CPP scores increased significantly at T1 compared with T0 (*n* = 12 per group). *** *p* < 0.001. (**B**): Heat maps showed differences in circRNAs expression in the nucleus accumbens (NAc) at T0 and T1 (*p* < 0.01). (**C**): The expression levels of circTmeff-1 and circDcun1d4 in the NAc on T0 and T1 (*n* = 5). *** *p* < 0.001. (**D**): The expression levels of circTmeff-1 and mRNA Tmeff-1 in the NAc after RNase R was administered, and the expression levels without RNase R were used as baseline (*n* = 5). *** *p* < 0.001. (**E**): CircTmeff-1 and GAPDH were amplified from cDNA and gDNA with divergent and convergent primers in the NAc cells of mice. (**F**): The RNA fluorescence in situ hybridization results demonstrated that the expression of circTmeff-1 (green) in T1 was increased compared with that in T0, and it could be seen that circTmeff-1 was mainly distributed in the cytoplasm. Scale bar = 50 μm.

**Figure 2 cells-12-01985-f002:**
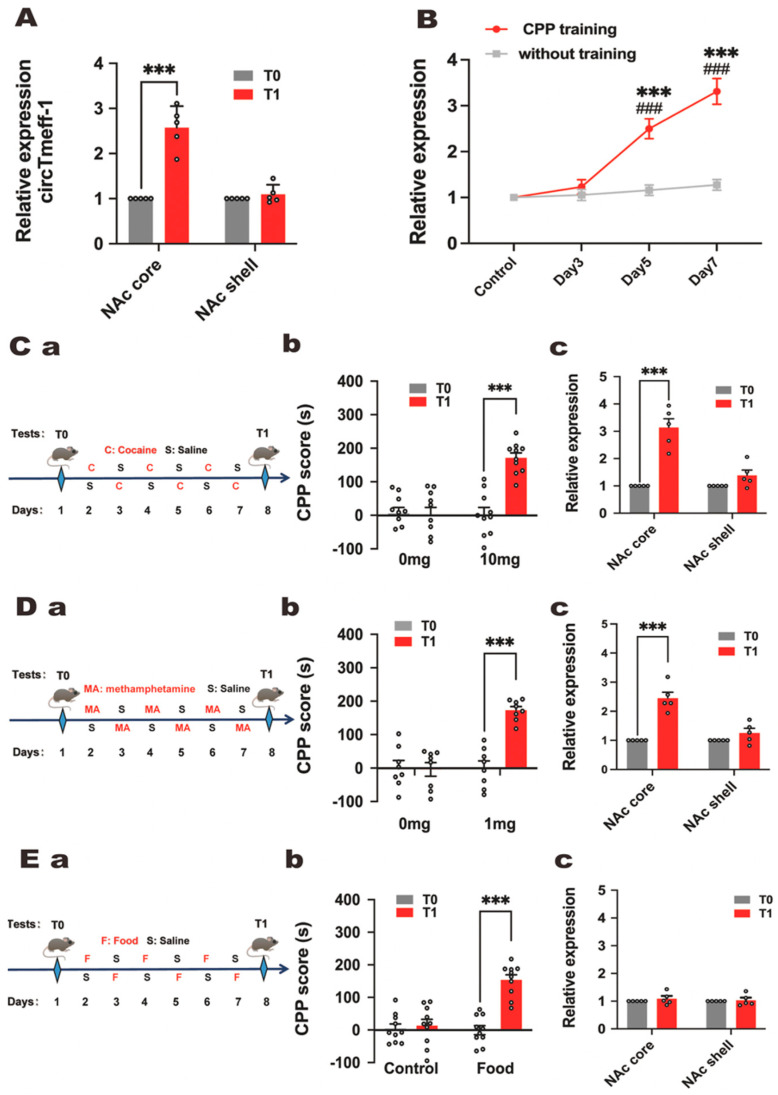
CircTmeff-1 was increased in the NAc of morphine CPP mice. (**A**): At T1, circTmeff-1 expression increased in the core of the NAc but not in the shell (*n* = 5 per group). *** *p* < 0.001. (**B**): circTmeff-1 increased gradually during CPP training in mice, whereas circTmeff-1 expression did not change after intraperitoneal injection of morphine alone without training (*n* = 5 per group). *** *p* < 0.001 vs. control group; ### *p* < 0.001 vs. without training group on the same day. (**C**): (**a**) Timeline diagram of cocaine CPP training paradigm. (**b**) CPP scores increased at T1 compared with T0 (*n* = 10 per group). *** *p* < 0.001. (**c**) At T1, circTmeff-1 in the NAc was increased in mice with cocaine (10 mg/kg) CPP training (*n* = 5 per group). *** *p* < 0.001. (**D**): (**a**) Timeline diagram of methamphetamine CPP training paradigm. (**b**) CPP scores increased at T1 (*n* = 8 per group). *** *p* < 0.001. (**c**) Increased circTmeff-1 in the NAc of mice with methamphetamine (1 mg/kg) CPP at T1 (*n* = 5 per group). *** *p* < 0.001. (**E**): (**a**) Timeline diagram of CPP training paradigm for high-sugar foods. (**b**) CPP scores increased at T1 (*n* = 10 per group). *** *p* < 0.001 (**c**) At T1, circTmeff-1 expression in the NAc of mice trained with high-glycemic diet CPP did not change (*n* = 5 per group).

**Figure 3 cells-12-01985-f003:**
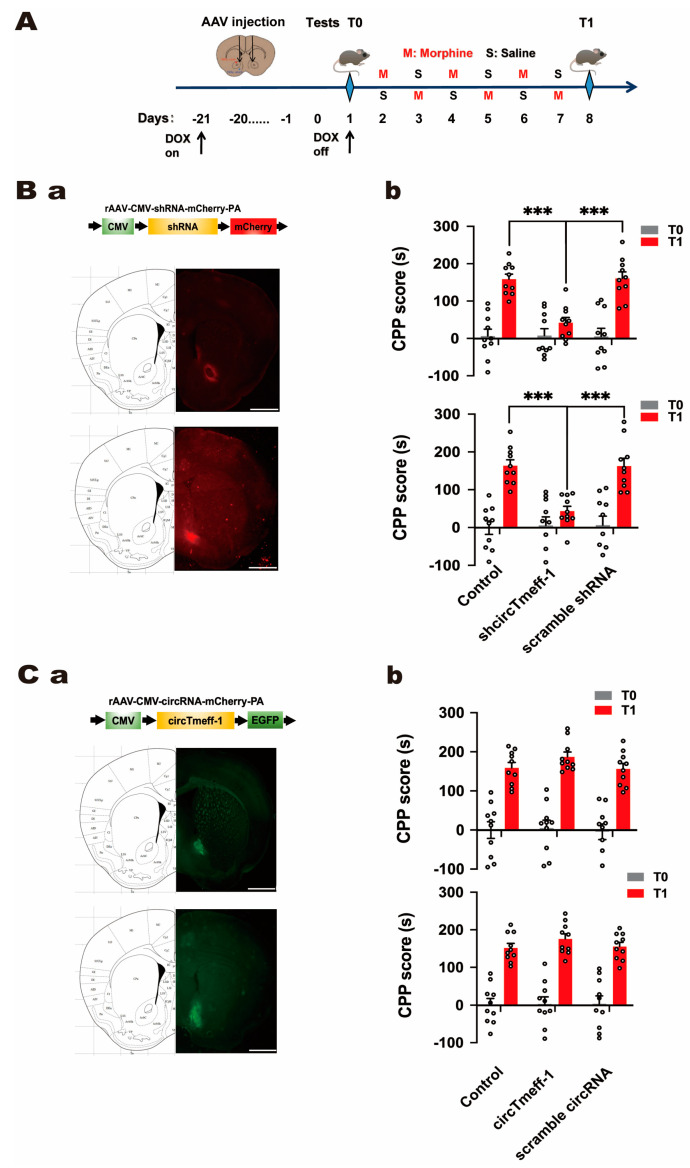
CircTmeff-1 modulates background-induced memory formation of morphine addiction. (**A**): a schematic diagram of the timeline of the morphine CPP training program. (**B**): Reducing circTmeff-1 expression in the NAc inhibits the expression of morphine addiction memory. (**a**) Schematic diagram of AAV structure used to down-regulate circTmeff-1; mCherry fluorescence location indicated that the injected virus was located in the core or shell of the NAc; scale bar = 1000 µm. (**b**) Down-regulation of circTmeff-1 in both NAc core and shell inhibited morphine seeking (*n* = 10 per group), *** *p* < 0.001. (**C**): Overexpression of circTmeff-1 in the core and shell of the NAc did not affect the expression of morphine-addicted memory. (**a**) Schematic diagram of AAV structure used for overexpression of circTmeff-1; EGFP fluorescence location indicated that the injected virus was located in the core or shell of NAc; scale bar = 1000 µm. (**b**) Overexpression of circTmeff-1 in the core and shell of the NAc had no effect on T1 background-induced morphine seeking (*n* = 10 per group). EGFP, enhanced green fluorescent protein.

**Figure 4 cells-12-01985-f004:**
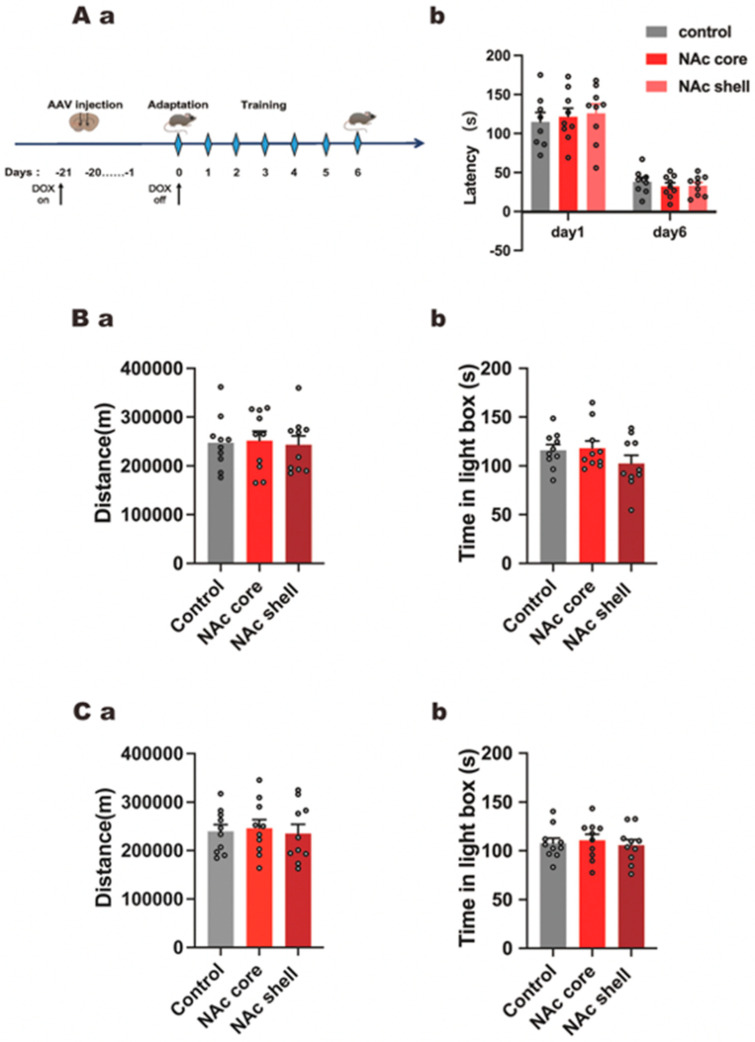
The variation in circTmeff-1 expressions by AAV did not affect motor activity, anxiety-like behavior, and spatial memory of mice. (**A**(**a**)): A schematic representation of the AAV injection and behavioral training timeline. (**A**(**b**)): Down-regulation of circTmeff-1 does not affect spatial memory induced by simultaneous bilateral NAc core or shell infusion of AAV (*n* = 8 per group). (**B**(**a**)): Down-regulation of circTmeff-1 in the NAc did not affect spontaneous activity in the mice (*n* = 10 per group). (**B**(**b**)): Down-regulation of circTmeff-1 in the core and shell of the NAc did not affect the time spent in the open chamber (*n* = 10 per group). (**C**(**a**)): Over-expression of circTmeff-1 in the NAc did not affect the spontaneous activity of mice (*n* = 10 per group). (**C**(**b**)): Over-expression of circTmeff-1 in the core and shell of the NAc did not affect the time spent in the open chamber (*n* = 10 per group). AAV, adeno-associated virus.

**Figure 5 cells-12-01985-f005:**
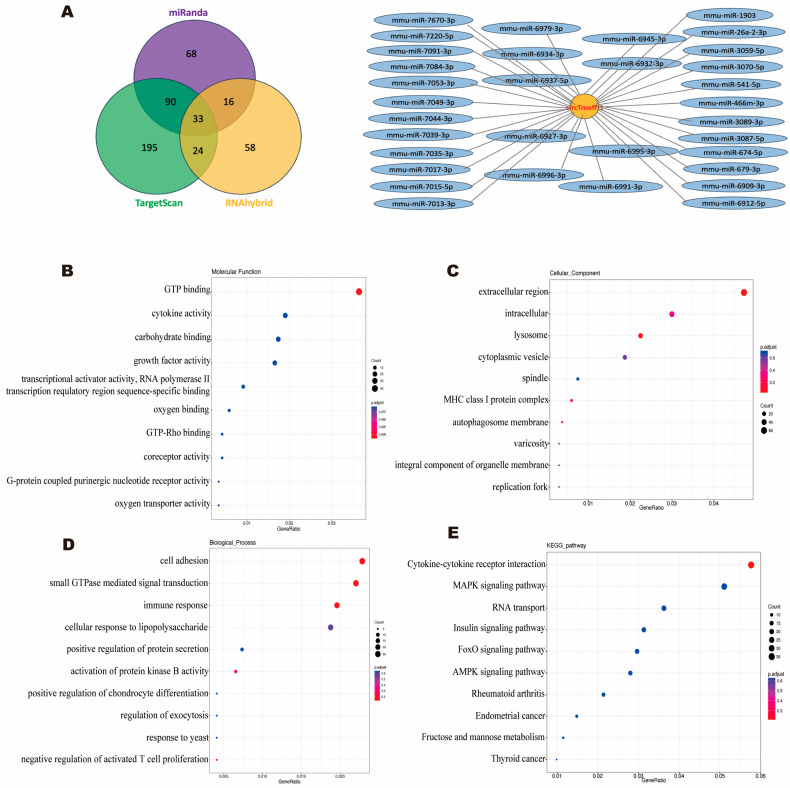
GO and KEGG analysis. (**A**): The prediction of downstream micro RNAs of circTmeff-1 using MiRanda, RNAhybrid, and TargetFinder tools. (**B**–**D**): Top 10 GO enrichment analysis, biological processes, cell component, and molecular function, in order. The x-coordinate represents GeneRatio, whereas the y-coordinate represents GO term. (**E**): Top10 enriched KEGG notes. Spot size represents the number of genes enriched, and blue to red represents the enrichment level in the corresponding item, from low to high.

**Table 1 cells-12-01985-t001:** Quantitative real-time PCR reaction conditions.

Step	Number of Cycles	Temperature	Time
Initial denaturation	1	95 °C	30 s
PCR reaction	40	95 °C 60 °C	5 s 30 s
Solution curve analysis	1	60–95 °C	2 min

**Table 2 cells-12-01985-t002:** The specific name of adeno-associated virus (AAV) and its titer.

Name	Titer
rAAV-TRE-5′-circRNA-nEfla-CMV-EGFP-PA	5.00 × 10^12^ vg mL^−1^
rAAV-TRE-5′-CMV-BSSI-nEfla-EGFP	2.86 × 10^12^ vg mL^−1^
rAAV-TRE-5′-mir30-shRNA(circRNA)-miR30-3′-CMV-mcherry-PA	8.23 × 10^12^ vg mL^−1^
rAAV-Tre3g-speI-5′-miR30a-xhoI-EcoRI-miR30a-3′-NheI-CMV-mcherry-PA	5.20 × 10^12^ vg mL^−1^
rAAV-CMV-tTA-WPRE-PA	6.93 × 10^12^ vg mL^−1^

## Data Availability

Data are available on request from the authors.

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
