# Peer review of "The Role of circTmeff-1 in Morphine Addiction Memory of Mice"

_cells, 2023, doi:10.3390/cells12151985_

Round 1

Reviewer 2 Report

Summary:

The manuscript by Hailei Yu et al. the authors do a thorough job of studying the involvement of the circRNA circTmeff-1 in primarily morphine induced addiction memory, through using the behavioral paradigm model CPP and expression analysis. In short, expression of circTmeff-1 is elevated in mice after CPP training, whilst knockdown of circTmeff-1 resulted in lower CPP scores, thus suggesting an involvement of circTmeff-1 in addiction memory and formation.

General concept comments:

It took me a moment to clearly understand that circTmeff-1 was primarily picked due to a previous story focusing on this specific circRNA. Could the authors clarify this more in the introduction.

In connection to the above. Why did the authors not include or also look at the other 4 circRNAs showing in figure 1B to have significant changes between T0 and T1? I think the story and especially the involvement of circRNAs in additive memory could potentially be strengthened by exploring both circRNAs that increase and decrease in expression after addiction memory. At least it would suit the manuscript to expand upon why these other circRNAs were not explored further.

The authors themselves mention in the discussion " Next, we hoped to detect proteins or miRNAs that may bind to circTmeff-1 and study the mechanism of circTmeff-1’s role in addiction by analyzing corresponding transcriptional genes". This is a bit of a broken sentence as it is since it is not really followed up by anything. However, why did the authors not investigate or look for expression of potential targets? The same purified RNA could be used to explore non-RNase R treated expression, which is not uncommon practice to ensure or distinguish true linear from circular RNA sequencing libraries.

I wonder why only 5 replicates or samples used for the qPCR, when in general more mice were examined in the behavioral tests. Why was not sample all mice included?  

Overall, the method section needs to be greatly expanded with necessary details for reproducibility. The RNA sequencing and qPCR section, for example, is lacking crucial details and will be hard to replicate. What protocols were used, amounts, kits, analytical pipelines, reference genome, annotation, qPCR anneal temperatures etc.

Specific Comments:

Schematics in Figure 1A and throughout can be a bit hard to interpret. Does it mean that 2 groups of mice altered between morphine and saline, or that the rooms changed? Please make it clearer for the reader, what it means.

Figure 1B: Missing axis of what other circRNAs showing the heatmap. Furthermore, legend explaining the color range.

Subfigure letters are confusing and redundant. Example Figure 2 Ca, Cb and Cc.

Line 251-252; Broken sentence.

Line 331 states " The present study identified that the escalation of circTmeff-1 in the NAc core and shell during the construction of the morphine-induced addiction memory model is crucial.". Unless I misunderstand, the study didn't find any escalation of circTmeff-1 in the shell (Figure 2).

Quality of English language is overall alright. There are several instances of broken sentences, or abruptly ending sentences. Similarly conclusion do not always match the results shown, see my other comments for more specifics.

Round 2

Reviewer 2 Report

I would like to thank the authors for the many clarifying comments and corrections. I am however still left with a few minor comments and details I would appreciate if the authors would accommodate. The points below are follow-up answers to the original comments/responses.

Point 1: Thanks for your comments.

Point 2. Not entirely sure if I agree with the authors overall comment, that the increase of circTmeff-1 is more pronounced. In their qPCR showing relative expression, yes, but in the RNA-seq the expression is similar. Was it done on the same RNA (qPCR and RNA-seq?) I am still missing a legend on the Figure1B explaining what the scale is for the heatmap. Please add it. I see in response to point 7, that the scale is TPM, please indicate it on the figure or in legend.

Regardless I would still very much like a small sentence from the results of Figure 1B+C that denotes the focus of this manuscript is purely on circTmeff-1, since as a reader, meaning me in this case, I was quite excited to also hear about the other ones found differently expressed in the RNA-seq data. I understand I will have to wait for the other manuscripts to appear, but for this stories sake, it would be nice to have an extra sentence here just clarifying: we will only focus on circTmeff-1 in this story, although we also find other interesting targets.

Point 3: Thanks for the clarification. Could you add the figure to the main manuscript since you now added a bit about it in methods as well? Otherwise, the information is quite unsubstantiated? Plus expand on how the miRanda, TargetScan, and RNAhybrid analysis was performed.

Point 4: Not entirely sure I understand the answer here. Why were only a selection of the mice tested in behavioral studies included in the gene expression test? Why not all? Was it a lack of material or some other reason?

Point 5: Thanks for the more detailed description. However, not to be pedantic, I would like a bit more information. For example, how was the RNA extracted for the RNA-seq analysis? Which RNA database was used to eliminate rRNA and how was the data filtered? Which annotation was used for the circRNAs?

Point 6: Thanks a lot for the clarification.

Point 7: Thanks for the explanation. Please add in figure or legend

Point 8: Thanks for the explanation, I will leave it up to the editors to decide if the subfigure labelling conforms to the journals format.

Point 9-11: Overall thanks for the response and clarifications.

No further comments on language.
